# Development of an Optimal Water Allocation Model for Reservoir System Operation

Eunkyung Lee [1] , Jungwon Ji [1], Seonmi Lee [1] , Jeongin Yoon [1], Sooyeon Yi [2] and Jaeeung Yi [1],*

1    Department of Civil Systems Engineering, Ajou University, 206 Worldcup-ro Yeongtong-gu, Suwon 16499, Republic of Korea; oplk100@ajou.ac.kr (E.L.); log58@ajou.ac.kr (J.J.); sunki7070@ajou.ac.kr (S.L.); pinkashley@ajou.ac.kr (J.Y.)
2    Landscape Architecture & Environmental Planning, University of California, Berkeley, CA 94720, USA; sooyeon@berkeley.edu
*    Correspondence: jeyi@ajou.ac.kr; Tel.: +82-31-219-2507

**Abstract:** Allocating adequate water supplies under the increasing frequency and severity of droughts is a challenge. This study develops an optimal reservoir system operation method to allocate water supplies from upstream reservoirs to meet the downstream water requirements; validates the proposed optimization model through the system operation of upstream reservoirs; and proposes new water supply policies that incorporate a transformed hydropower reservoir with an add-on water supply function and two multipurpose reservoirs. We use linear programming to develop an optimal water allocation model. This model provides an operational strategy for managing upstream reservoirs with different storage capacities. By integrating the effective storage ratio of each reservoir into the allocation estimation, the model ensures an optimal distribution of downstream water requirements. The results indicated well-balanced, effective storage ratios among the Chungju, Soyanggang, and Hwacheon Reservoirs across varying hydrological conditions. Specifically, during drought years, the average effective storage rates were 20.5%, 20.6%, and 19.07%, respectively. In normal years, these figures, respectively, were 59.3%, 68.6%, and 52.4%, while in wet years, the rates stood at 64.08%, 62.90%, and 54.61%. This study enriches the reservoir operation literature by offering adaptable solutions for collaborative reservoir management and presents efficient strategies for reservoir operations.

**Keywords:** water allocation model; reservoir optimization; effective storage ratio; linear programming; water resources management; Han River Basin

## 1. Introduction

Climate change is significantly altering hydrologic systems, intensifying precipitation in wetter regions while exacerbating drought in drier areas [1,2]. Extreme events like droughts pose severe challenges to both water supply and demand [3]. South Korea has an average annual precipitation of 1331.7 mm, which is well above the global rainfall average of 884 mm [4]. South Korea experiences four distinct seasons, resulting in substantial changes in precipitation and temperature. Approximately 68% of its annual precipitation occurs during the flood season from June to September, and precipitation varies greatly temporally and spatially [5]. With 65% of its terrain being mountainous with slopes exceeding 20%, the country experiences rapid run-off into rivers [6]. To manage its water resources, South Korea relies extensively on reservoirs throughout the country [7]. In the event of a water deficit during the dry season, reservoirs supply the stored water during the flood season to prevent drought. However, challenges arise during periods of insufficient rainfall throughout the year. The country experiences long-term drought cycles of 5–7 years and short-term cycles of 2–3 years [8]. Localized droughts have become more frequent in recent years, contributing to increased water scarcity. Due to a lack of precipitation throughout the summer months, a prolonged drought occurred from 2014 to 2017. The

drought during this period is considered the most severe in the Han River Basin to date, and long-term droughts are anticipated to worsen as a result of climate change [9,10].

Preparing for drought often involves increasing storage capacity, commonly achieved through the construction of new reservoirs [11]. However, these construction projects come with notable environmental and social drawbacks [12]. Social conflicts can emerge between upstream communities, potentially affected by submersion from a new reservoir, and downstream communities, who may experience alterations in flow regimes, water quality, and ecosystems [13]. Also, changing reservoir operations can affect the natural streamflow regime and impact the aquatic ecosystem biodiversity. The construction of reservoirs, as exemplified by the Lake Powell River Reservoir in the Colorado River, USA, modifies the natural river flow [14]. The building and functioning of the Three Gorges Dam have substantially changed the hydrological patterns downstream on the Yangtze River, impacting environmental conditions, biodiversity, landscape structure, and human development [15]. Recognizing these trade-offs, the subsequent sections of this study focus on optimizing reservoir operations to balance these competing interests. To address water scarcity, the South Korean government has investigated ways to secure additional water supplies, for example, by adding on a water supply function to the existing hydropower reservoirs in the Han River Basin. The Soyanggang and Chungju Reservoirs supply water to Seoul, the capital city of South Korea. In April 2020, the Korean government decided to add a water supply function to the Hwacheon Reservoir, which originally served as a hydropower reservoir, to better meet the increasing water demand upon severe droughts [16].

Numerous countries leverage existing reservoirs to fulfill their water requirements. An initiative movement for the multifunctional use of existing hydropower reservoirs emerged prominently during the 6th World Water Forum in Marseille, France, in 2012 led by Électricité de France and the World Water Council [17]. Many countries, like Peru, Costa Rica, the United States, France, Cameroon, Niger, India, Nepal, China, and Austria, have already added water supply functions to the existing hydropower reservoirs [18]. For example, the Serre-Ponçon Reservoir in France primarily served as hydropower generation but now supports secondary functions like water supply and local tourism [19]. Extensive research has also been conducted on the multipurpose usage of hydropower reservoirs for water demand and ecosystem preservation. As another example, the Keswick Reservoir, a hydropower reservoir in California, USA, added a water supply function to better meet the domestic and ecosystem water needs by developing the optimal reservoir operation rules [20].

Although initially constructed for hydropower generation, the Hwacheon Reservoir has substantial storage capacity and is a desirable candidate to potentially have a water supply function. In the past, the Hwacheon Reservoir irregularly released water during some drought events. However, these operations have often proceeded without a systematic framework. The Hwacheon Reservoir in South Korea stands as the first example of a transformed multipurpose reservoir that encompasses not only power generation but also water supply. However, the purposes of water supply and hydropower generation conflict with each other occasionally. This is because the timing of hydropower generation is not always compatible with the timing of water supply.

Several studies have aimed to estimate the volume of water that the Hwacheon Reservoir could provide with a 95% reliability for water supply. This initiative led to the development of a Hwacheon Reservoir rule curve specifically for drought conditions [21]. Another study proposed estimating the water supply capacity for an individual reservoir by incorporating operational aspects of hydropower reservoirs into a reservoir operation model [22,23]. Another study used a data-driven model for predicting inflows into hydropower reservoirs and assessed the water supply capabilities of hydropower reservoirs [24]. Prior research has predominantly focused on operation of a single reservoir, leaving a gap in the literature regarding integrated operations with other reservoirs for effective water resources management. In reviewing the existing literature, several gaps be-

come evident: While there are instances in other countries of hydropower reservoirs being converted for multipurpose use, there is an absence of research focusing on the cooperative operation of such transformed reservoirs with other types of reservoirs. Previous research has largely centered on securing water supply through the cooperative management of multipurpose and water-only reservoirs. Studies have also been conducted on reservoir simulation considering emergency storage and water supply volumes within the same basin. Notably, there is a lack of studies applying optimization techniques to efficiently allocate water through the cooperative operation of existing multipurpose reservoirs initially designed for water supply with hydropower reservoirs newly repurposed for water supply. These gaps in the literature underscore the novelty and significance of the present study, which aims to address these unexplored areas.

To bridge this gap, the present study aims to achieve multiple objectives to enhance water resource management in the Han River Basin through a more integrated approach. Specifically, this study seeks to (a) develop an optimal reservoir system operation method to allocate water supplies from multiple upstream reservoirs to meet the downstream water requirements; (b) validate the proposed optimization model through the system operation of three upstream reservoirs using historical inflow data; and (c) propose new water supply policies that incorporate a transformed hydropower reservoir with an add-on water supply function and two multipurpose reservoirs. This ensures an optimal downstream water supply while accounting for the unique characteristics and status of the upstream reservoirs.

The novel contributions of this study are multifold: Unlike traditional studies that focus solely on multipurpose reservoirs, our work extends the functionality of a hydropower reservoir by incorporating a water supply component. Our research addresses a gap in current methodologies by developing rules for water allocation, particularly when only a limited number of reservoirs are available to meet downstream demand. The paper presents a case study that transforms an existing hydropower reservoir to serve additional functions, offering a cost-effective and environmentally sustainable alternative to building new infrastructure. This study offers a collaborative framework for cases like South Korea, where multipurpose reservoirs and their operating institutions differ, thereby necessitating cooperation for optimal reservoir management. This holds true even if the same institution operates all the reservoirs, enabling rational operations to achieve the best results in non-flood water supply seasons. In the context of South Korea, the transformation of existing multipurpose reservoirs by adding hydropower functions for water supply enables the securing of additional water volumes, thereby contributing to the optimal integrated operation of the three reservoirs to prevent water supply shortages during non-flood seasons. These contributions bring new perspectives to the field of water resources management and offer practical solutions for enhanced system operation.

## 2. Water Allocation Methods for Multireservoir Systems

Optimization models serve as invaluable tools in water resources management for designing optimal system configurations or operational measures [25,26]. Managing reservoirs is complex, requiring a balance of diverse operational objectives for optimal operation [27]. Numerous studies have employed optimization models to develop and assess reservoir operations. Various optimization techniques, such as linear programming (LP) [28,29], nonlinear programming (NLP) [30,31], dynamic programming (DP) [32,33], and heuristics algorithms (HA) [34–36], have been explored for over four decades. These models aim to either maximize or minimize reservoir objectives while satisfying operational constraints [37]. A crucial step in model development is the careful formulation of the objective function and constraints. The model in this study employs LP, a commonly used technique in the water resources field for its simplicity and capability to find the global optimum [38].

The water level within a reservoir fluctuates from a low water level to either a normal high or a restricted level during flood seasons. The release of water is based not just on

downstream demand, but also on reservoir storage conditions [39–41]. Reservoir system operation considers conditions of the individual reservoir such as inflows, storages, and demands, enabling more effective water resources management. Consequently, the topic of reservoir system operations in the basin has long been a subject of ongoing research, as it provides a more stable water supply and better flood management compared to single reservoir operations [42]. Earlier studies have developed guidelines for the operation of single-purpose reservoirs arranged in series or parallel. Further research has categorized reservoir system operation into those focused on water supply and flood control, leading to more efficient operation plans depending on the objective [43–45]. Specifically, various approaches have been investigated to determine the supply allocation from multiple reservoirs to meet downstream demand [46].

### 2.1. Method 1: Dry Season Allocation Using Predicted Inflow and Available Reservoir Storage

Effective storage in a reservoir is the storage capacity between the low water level and normal high water level. Available storage is the storage capacity between the current water level and the normal high water level, which is reserved space for additional water storage (Equation (1)). The first step is to determine the proportion of available storage in each reservoir relative to the total available storage across all reservoirs (Equation (2)). The second step is to estimate the predicted inflow for the dry season (from the present to the onset of the flood season) for each reservoir. The third step is to assess the proportion of each reservoir's predicted inflow to the total predicted inflow for the dry season (Equation (3)). The final step involves calculating each reservoir's allocation, which is the ratio of its available storage to the predicted inflow during the dry season (Equation (4)).

This method is only applicable to the period preceding the flood season. In South Korea, the annual cycle is divided into a flood season (21 June–20 September) and a dry season (21 September–20 June of the following year). The water allocation for each reservoir is determined to secure available storage. This allocation ensures a consistent ratio of inflow in the dry season to available storage for all reservoirs (Equation (5)).

The total demand is equal to the sum of the allocation amounts for the reservoirs in a given month (Equation (6)). In Equation (5), setting $\sum_{i=1}^{n}[SN_{ii} - (S_{i,t-1} + i_{i,t})]$ equal to $N_t$ and integrating with Equation (6) yields Equation (7). The allocation amount for each reservoir is expressed in Equation 8.

This approach tends to minimize reservoir release during the dry season because it only considers relative available storage. However, multipurpose reservoirs in South Korea need to supply more than the contracted amount of water. This method does not account for the minimum release requirements for each reservoir and is uncertain when estimating the allocation amount in flood season.

$$v_{i,t} = SN_{i} - (S_{i,t-1} + i_{i,t} - x_{i,t}) \tag{1}$$

where $v_{i,t}$ is the available storage in reservoir $i$ for the period $t$; $SN_i$ is the reservoir storage at the normal high water level during the dry season and the restricted water level during flood season; $S_{i,t}$ is the storage of reservoir $i$ for the period $t$; $i_{i,t}$ is the inflow of reservoir $i$ for the period $t$; and $x_{i,t}$ is the allocation from reservoir $i$ for the period $t$.

$$P_{i,t} = \frac{v_{i,t}}{\sum_{i=1}^{n} v_{i,t}} = \frac{SN_{i} - (S_{i,t-1} + i_{i,t} - x_{i,t})}{\sum_{i=1}^{n}[SN_{i} - (S_{i,t-1} + i_{i,t} - x_{i,t})]} \tag{2}$$

$$\alpha_{i,t} = \frac{ir_{i,t}}{\sum_{i=1}^{n} ir_{i,t}} \tag{3}$$

where $ir_{i,t}$ is the predicted inflow into reservoir $i$ during the remainder before the flood season.

$$P_{i,t} = \alpha_{i,t} \tag{4}$$

where $P_{i,t}$ is the ratio of free volume in reservoir $i$ to the total free volume in the system, and $\alpha_{i,t}$ is the proportion of the period $t$'s demand allocated to reservoir $i$.

$$\alpha_{i,t} = \frac{SN_i - (S_{i,t-1} + i_{i,t} - x_{i,t})}{\sum_{i=1}^{n}[SN_i - (S_{i,t-1} + i_{i,t} - x_{i,t})]} \tag{5}$$

$$\sum_{i=1}^{n} x_{i,t} = D_t \tag{6}$$

where $D_t$ is the total demand.

$$SN_i - (S_{i,t-1} + i_{i,t} - x_{i,t}) = \alpha_{i,t}(N_t + D_t) \tag{7}$$

where $N_t$ is the reservoir's available storage, excluding the portion occupied by the current water storage from full storage capacity.

$$x_{i,t} = \alpha_{i,t}(N + D) - SN_i + S_{i,t-1} + i_{i,t} \tag{8}$$

*2.2. Method 2: Allocation Based on Storage Ratio*

This method estimates the allocation amount for this month using the storage ratio at the end of the preceding month for each reservoir (Equation (9)). A higher storage ratio results in a larger allocation amount. This method only considers storage, and it does not consider future reservoir condition, capacity, and inflow. Thus, this method is not applicable in real reservoir operation.

$$x_{i,t} = \left[\frac{S_{i,t-1}}{\sum_{i=1}^{n} S_{i,t-1}}\right] \times D_t \tag{9}$$

*2.3. Method 3: Allocation Considering Storage and Inflow*

This method considers both storage and predicted inflow to determine the water allocation amount (Equation (10)). The allocation amount is estimated based on the ratio of available water, which is computed by adding reservoir storage for the previous month and the predicted inflow for the current month. This approach is suitable for reservoirs with similar capacities but not for those with significant differences in reservoir capacities. This method is impractical, as it relies only on the volume of available water while neglecting the hydrologic conditions and capacity of each reservoir.

$$x_{i,t} = \left[\frac{S_{i,t-1} + i_{i,t}}{\sum_{i=1}^{n}(S_{i,t-1} + i_{i,t})}\right] \times D_t \tag{10}$$

*2.4. Method 4: Allocation Using Effective Storage Ratio*

The method incorporates the effective storage ratio of each reservoir in allocation estimation (Equation (11)). The effective storage is achieved by subtracting the storage at the low water level from the storage at the normal high water level. The current effective storage is calculated by subtracting the storage at the low water level from the storage at the current water level. The effective storage ratio is the current effective storage divided by the effective storage. This method can estimate the allocation amount for those reservoirs with different storage capacities (Equation (12)). Thus, this study adopts this method, as it considers both reservoirs with different capacities and the current reservoir condition.

$$R_{i,t} = \left[\frac{S_{i,t} - SL_i}{SN_i - SL_i}\right] \times 100 \tag{11}$$

$$x_{i,t} = \left[\frac{R_i}{\sum_{i=1}^{n} R_i}\right] \times D_t \tag{12}$$

where $R_{i,t}$ is the effective reservoir storage rate in reservoir $i$ for the period $t$, and $SL_i$ is the reservoir storage at the low water level in reservoir $i$.

### 2.5. Optimization Model for Equitable Water Allocation in Multireservoir Systems

In this study, we develop an optimization model for water allocation using the effective storage ratio method. This method provides an optimal operational strategy for downstream water requirements. By integrating the effective storage ratio of each reservoir into the allocation estimation, the model ensures an optimal distribution of downstream water requirements while considering the different reservoir capacities and the current reservoir condition. The objective of the optimization model is to maximize the sum of the minimum monthly storage ratios for each reservoir throughout its operation period (Equation (13)). This objective function is different from its traditional objective function, which maximizes the sum of the monthly storage for all reservoirs. Such a traditional approach can lead to an operation where a single reservoir supplies all downstream demands, increasing the storage of other reservoirs. This study uses an objective function that maintains a similar effective storage ratio for all reservoirs.

$$\max \sum_{t=1}^{12} R_{i_{min},t} \tag{13}$$

where $R_{i_{min},t}$ is the water storage rate for reservoir $i$ with a minimum $R$ among reservoirs in month $t$.

Constraints of the optimization models are as follows. Equation (14) represents the water balance equation in a reservoir. The sum of all allocations from upstream reservoirs should exceed the downstream water requirement (Equation (15)). For each reservoir, the effective storage ratio is estimated using (Equation (16)) and is required to surpass the minimum storage ratio for all reservoirs (Equation (17)). The storage for each reservoir should remain between the storage for the low water level and the storage for the normal high water level (Equation (18)). The allocation amount from each reservoir should exceed the planned water supply or the minimum instream flow (Equation (19)). The planned water supply is the predetermined volumes of water released from the multipurpose reservoirs. All variables are constrained to be positive (Equation (20)).

$$S_{i,t} = S_{i,t-1} + i_{i,t} - x_{i,t} - w_{i,t} \tag{14}$$

where $w_{i,t}$ is the discharge excluding the allocation of reservoir $i$ in month $t$.

$$x_{1,t} + x_{2,t} + \cdots + x_{n,t} \geq D_t \tag{15}$$

$$R_{i,t} = \left[ \frac{S_{i,t} - SL_i}{SN_i - SL_i} \right] \times 100 \tag{16}$$

$$R_{i,t} \geq R_{i_{min},t} \tag{17}$$

$$SL_i \leq S_{i,t} \leq SN_i \tag{18}$$

$$x_{i,t} \geq IF_i \text{ or } WSP_{i,t} \tag{19}$$

$IF_i$ represents the stream maintenance flow of reservoir $i$; $WSP_{i,t}$ is the planned water supply of reservoir $i$ for month $t$; and $n$ is the number of reservoirs.

$$S_{i,t}, \, i_{i,t}, \, x_{i,t}, \, w_{i,t}, \, SL_i, \, SN_i, \, R_{i,t}, \, R_{min,t}, \, IF_i, \, WSP_{i,t} \geq 0 \tag{20}$$

## 3. Application

### 3.1. Study Area

The capital area of South Korea, encompassing Seoul, Incheon Metropolitan City, and Gyeonggi Province, covers an area of 11,856 km$^2$, or 11.8% of the nation's total land area of 100,210 km$^2$. This region has approximately 26 million people, which accounts for 50.5% of the national population [47]. Due to its high population density, the capital area is sensitive to water supply shortage. With four distinct seasons with the most precipitation in the

summer, reservoirs are crucial for storing water during flood season for later use in the dry season.

The Han River system includes three multipurpose reservoirs (Soyanggang, Chungju, and Hoengseong), seven hydropower reservoirs (Hwacheon, Chuncheon, Uiam, Cheongpyeong, Goesan, Paldang, and Doam), and one flood control reservoir (Pyeonghwa) (Figure 1 and Table 1). The three multipurpose reservoirs release predetermined volumes of water (Table 2). The Hoengseong Reservoir, completed in 2000, is designed for water scarcity mitigation and flood control in the Seomgang River Basin, a Han River tributary. The Soyanggang and Chungju Reservoirs supply water and control floods in the capital area. Soyanggang Reservoir is the largest reservoir and was completed in 1973 with a basin area of 2703 km². The Chungju Reservoir, built in 1985, has a basin area of 6648 km² with an average annual inflow of 154.5 CMS—approximately 2.78 times greater than the inflow for Soyanggang (55.5 CMS). The effective storage of Hoengseong Reservoir is relatively small and only accounts for 3.00% and 3.16% of the storage capacities of Soyanggang and Chungju, respectively. The Hoengseong Reservoir is a substantially smaller multipurpose reservoir that only accounts for 8.92% of the capacity of the Hwacheon Reservoir.

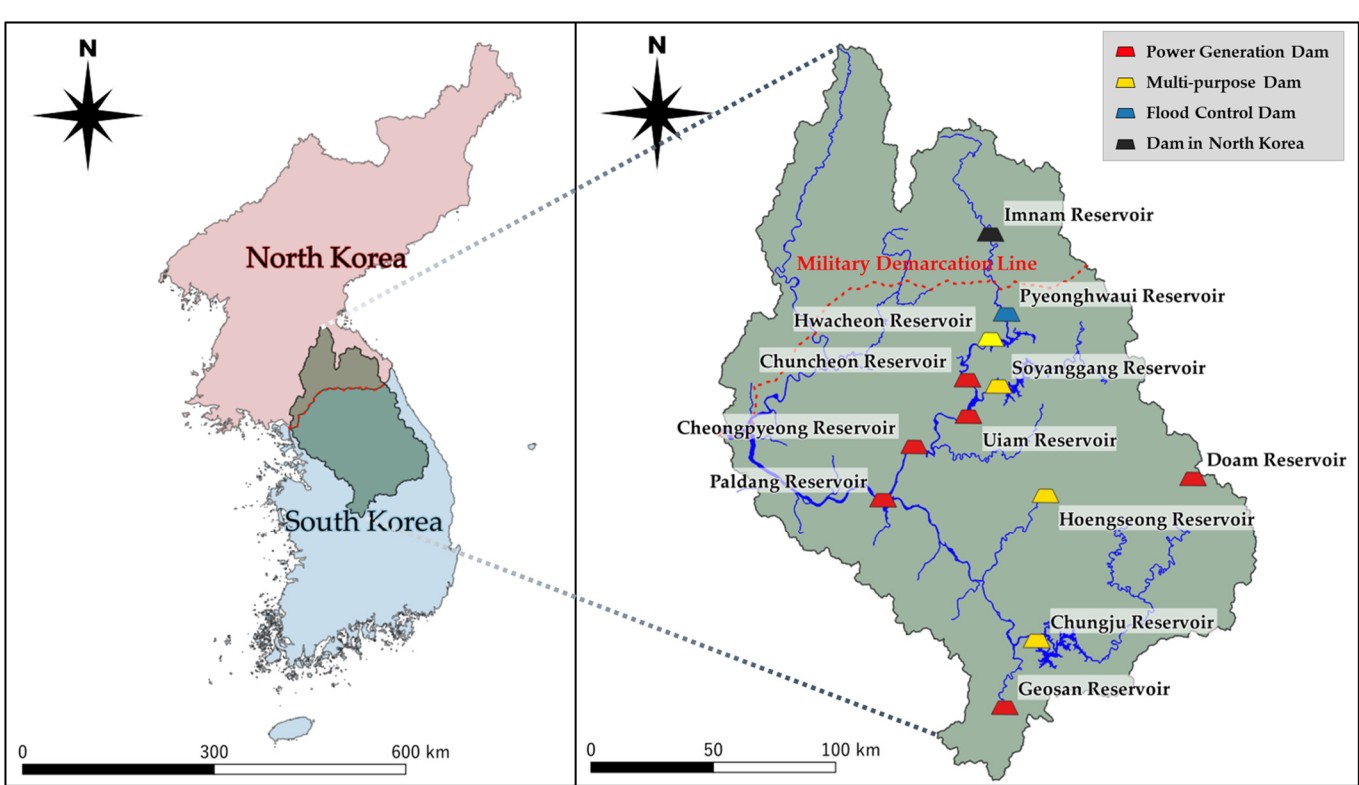

**Figure 1.** A study area map of Han River Basin.

**Table 1.** Comparative analysis of storage capacities (million cubic meters (MCM)) and water levels (EL.m) in multipurpose and Hwacheon reservoirs.

| Reservoir | Total Storage | Conservation Storage | Flood Water Level | Normal High Water Level | Restricted Water Level | Low Water Level |
|---|---|---|---|---|---|---|
| | MCM | MCM | EL.m | EL.m | EL.m | EL.m |
| Chungju | 2750 | 1789 | 145 | 141 | 138 | 110 |
| Soyanggang | 2900 | 1900 | 198 | 193.5 | 190.3 | 150 |
| Hoengseong | 86.9 | 73.4 | 180 | 180 | 178.2 | 160 |
| Paldang | 244 | 18 | 27 | 25.5 | na | 25 |
| Hwacheon | 974.2 | 572.8 | 183 | 181 | 175 | 156.8 |

**Table 2.** Monthly planned water supply (WS) of the multipurpose reservoirs. Multipurpose reservoirs serve as domestic (D), industrial (I), agricultural (A), and instream (IS) water supply (CMS). A* is the water intake within the reservoir, and B** is the water intake from downstream of the reservoir.

| Reservoir | | Soyanggang | | | Chungju | | | Hoengseong | | |
|---|---|---|---|---|---|---|---|---|---|---|
| Types of WS | | D and I | Ag | IS | D and I | Ag | IS | D and I | Ag | IS |
| January | A* | - | - | - | - | - | - | 2.3 | - | - |
| | B** | 38.1 | - | 8.1 | 86.6 | - | 10.6 | - | - | 1.8 |
| February | A* | - | - | - | - | - | - | 2.3 | - | - |
| | B** | 38.1 | - | 8.1 | 86.6 | - | 10.6 | - | - | 1.5 |
| March | A* | - | 0.4 | - | - | - | - | 2.3 | - | - |
| | B** | 38.1 | - | 8.1 | 86.6 | - | 10.6 | - | - | 0.9 |
| April | A* | - | 1 | - | - | - | - | 2.3 | 0.1 | - |
| | B** | 38.1 | - | 8.1 | 86.6 | 9.1 | 10.6 | - | 0.3 | 0.5 |
| May | A* | - | 1 | - | - | - | - | 2.3 | 0.1 | - |
| | B** | 38.1 | - | 8.1 | 86.6 | 21.8 | 10.6 | - | 1 | 1.8 |
| June | A* | - | 1 | - | - | - | - | 2.3 | 0.1 | - |
| | B** | 38.1 | - | 8.1 | 86.6 | 28 | 10.6 | - | 1 | 2.1 |
| July | A* | - | 1 | - | - | - | - | 2.3 | 0.1 | - |
| | B** | 38.1 | - | 8.1 | 86.6 | 18 | 10.6 | - | 1 | 0.5 |
| August | A* | - | 1 | - | - | - | - | 2.3 | 0.1 | - |
| | B** | 38.1 | - | 8.1 | 86.6 | 23.7 | 10.6 | - | 1 | - |
| September | A* | - | 1 | - | - | - | - | 2.3 | 0.1 | - |
| | B** | 38.1 | - | 8.1 | 86.6 | 10.6 | 10.6 | - | 0.7 | 0.4 |
| October | A* | - | 0.3 | - | - | - | - | 2.3 | 0.1 | - |
| | B** | 38.1 | - | 8.1 | 86.6 | 8 | 10.6 | - | 0.2 | 0.2 |
| November | A* | - | - | - | - | - | - | 2.3 | 0.1 | - |
| | B** | 38.1 | - | 8.1 | 86.6 | - | 10.6 | - | 0.1 | 1.2 |
| December | A* | - | - | - | - | - | - | 2.3 | - | - |
| | B** | 38.1 | - | 8.1 | 86.6 | - | 10.6 | - | - | 1.3 |
| Mean | A* | - | 0.4 | - | - | - | - | 2.3 | 0.1 | - |
| | B** | 38.1 | - | 8.1 | 86.6 | 10 | 10.6 | - | 0.4 | 1 |

Reservoir management is divided among different agencies. For example, K-water manages the multipurpose reservoirs and the Korea Hydro & Nuclear Power Company operates hydropower reservoirs. While flood control and hydropower reservoirs serve specific functions, multipurpose reservoirs are versatile, catering to water supply, flood control, and energy generation. The importance of securing a consistent water supply for the capital area was highlighted during the severe drought from 2015 to 2018, prompting the government to explore expanding the roles of multipurpose reservoirs within the Han River system. In April 2020, a pilot project commenced to convert the Hwacheon Reservoir, the largest hydropower reservoir in the Han River system, into a multipurpose reservoir. This marks the first conversion of a hydropower reservoir to a multipurpose reservoir in South Korea. Since then, the Hwacheon Reservoir constantly releases 22.2 CMS.

This study aims to develop an optimization model for the coordinated operation of the Hwacheon Reservoir and the Chungju and Soyanggang Reservoirs. Among the hydropower reservoirs, this study considers the Hwacheon and Paldang Reservoirs, while other hydropower reservoirs are excluded because they are run-off reservoirs. Among the multipurpose reservoirs, this study considers the Soyanggang and Chungju Reservoirs. The Hoengseong Reservoir is excluded because it is a local reservoir with a small capacity. The multipurpose and hydropower reservoirs in the Han River system are operated jointly only during emergencies, such as drought or floods, under the supervision of the Han River Flood Control Center. For example, if the Chungju, Soyanggang, and Hwacheon Reservoirs are operated by a single agency, then this agency can consider the specific conditions of each reservoir for all.

### 3.2. Data Collection

North and South Korea share the North Han River. The Imnam Reservoir in North Korea is located upstream on the North Han River, while the Hwacheon Reservoir in South Korea is downstream of Imnam Reservoir. Built in 2004, the Imnam Reservoir diverts inflow into the East Sea year-round for hydropower generation. During the flood season, Imnam Reservoir releases water via its spillway, leading to downstream flooding events in South Korea. Conversely, the release from the reservoir diminishes during the dry season. Since the construction of the Imnam Reservoir, the average inflow into the Hwacheon Reservoir has notably decreased from 90.1 CMS (based on average data from 1967 to 2003) to 51.3 CMS (based on average data from 2004 to 2022), representing a 57.0% reduction (Figure 2).

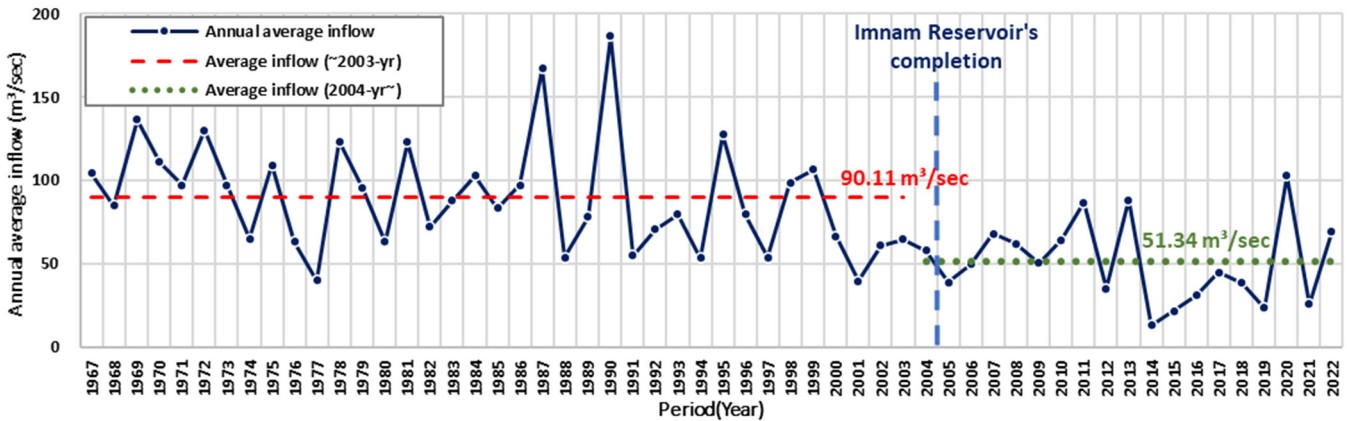

**Figure 2.** Annual average inflow to Hwacheon Reservoir, 1967–2022: comparison of pre- and post-2004 periods.

This study used historical inflow data, focusing on the period after 2004 when the inflow of the Hwacheon reservoir was significantly reduced upon the completion of Imnam Reservoir. We built and tested the model for every single year from 2004 to 2022. However, instead of presenting the results for a total of 19 years' worth of data, we identified three representative years for dry, normal, and wet conditions. Between 2004 and 2022, the average inflows into the Chungju, Soyanggang, and Hwacheon Reservoirs were 150.1 CMS, 67.5 CMS, and 51.3 CMS, respectively (Table 3). The year 2015 was selected as the dry year because it experienced an extended drought and had the lowest average annual inflow. The year 2018 was selected as a normal year, as its annual average inflow value was closest to the average inflow value. The year 2020 was chosen as a wet year due to its highest average annual inflow. We applied the monthly inflow data for 2015, 2018, and 2020 to develop the model and to assess the feasibility of system operation for the three reservoirs. All input data came from the Han River Flood Control Office (http://www.wamis.go.kr/, accessed on 13 April 2023).

**Table 3.** Average annual inflow (CMS) for Chungju, Soyanggang, and Hwacheon Reservoirs from 2004 to 2022.

| Year | Chungju Reservoir (CMS) | Soyanggang Reservoir (CMS) | Hwacheon Reservoir (CMS) |
|---|---|---|---|
| 2004 | 214.0 | 81.1 | 58.0 |
| 2005 | 175.4 | 63.1 | 39.0 |
| 2006 | 244.7 | 95.8 | 49.9 |
| 2007 | 212.2 | 76.2 | 68.1 |
| 2008 | 96.3 | 58.6 | 62.3 |
| 2009 | 128.0 | 75.3 | 50.8 |

**Table 3.** *Cont.*

| Year | Chungju Reservoir (CMS) | Soyanggang Reservoir (CMS) | Hwacheon Reservoir (CMS) |
|---|---|---|---|
| 2010 | 169.0 | 74.8 | 64.4 |
| 2011 | 283.1 | 105.2 | 86.9 |
| 2012 | 159.7 | 56.0 | 35.5 |
| 2013 | 144.8 | 75.1 | 88.5 |
| 2014 | 73.5 | 29.2 | 13.3 |
| 2015 | 55.6 | 33.8 | 21.7 |
| 2016 | 91.7 | 52.1 | 31.4 |
| 2017 | 108.6 | 62.6 | 45.1 |
| 2018 | 160.7 | 66.8 | 38.5 |
| 2019 | 74.7 | 36.4 | 23.8 |
| 2020 | 193.9 | 107.0 | 102.9 |
| 2021 | 108.6 | 45.3 | 26.1 |
| 2022 | 167.8 | 87.3 | 69.3 |
| Average | 150.1 | 67.5 | 51.3 |

## 4. Results

This study developed a model to allocate appropriate water supplies from the Chungju, Hwacheon, and Soyanggang Reservoirs to meet the required discharge for Paldang Reservoir (124 CMS and 138 CMS for the nursery and transplantation season). The model incorporated the specialized discharge requirements for Paldang Reservoir during the nursery and transplantation period from 27 May to 10 June, with rates set at 126.3 CMS for May and 128.7 CMS for June. The effectiveness of the model was evaluated using actual inflow data from three representative years: a dry year (2015), a normal year (2018), and a wet year (2020).

### 4.1. Model Evaluation for Dry Year (2015)

Supplementary Tables S1 (Chungju Reservoir), S2 (Soyanggang Reservoir) and S3 (Hwacheon Reservoir) present the outcomes of applying the optimization model during the drought year of 2015. These tables delineate the optimal water allocation for each reservoir with the spillway discharge. These tables are represented in Figure 3a–c.

In December 2014, the initial water levels for the three reservoirs (end-of-month water levels) were as follows: 126.2 EL.m for Chungju Reservoir, 165.8 EL.m for Soyanggang Reservoir, and 165.2 EL.m for Hwacheon Reservoir. The continuing drought since 2014 resulted in low initial water levels for reservoir operations in 2015. Given the modest inflows and low water levels during the dry season of 2015, the model requirement for each reservoir to release more than the planned water supply was unsatisfied. Thus, only instream flows were discharged during the dry season.

In June, the reservoirs recorded their lowest water levels: 114.5 EL.m for Chungju Reservoir, 156.3 EL.m for Soyanggang Reservoir, and 159.1 EL.m for Hwacheon Reservoir (Tables 4–6). Due to their different basin areas and locations, these reservoirs showed significant differences in average annual inflows. Hence, the average annual inflows for the Chungju, Soyanggang, and Hwacheon Reservoirs were 55.4 CMS, 33.6 CMS, and 21.5 CMS, respectively. Their average annual allocation amount for historical data was 65.8 CMS (52.8%), 36.1 CMS (29.0%), and 22.7 CMS (18.2%), respectively. Among the three reservoirs, the Chungju reservoir had larger initial storage than the other two and received a large initial allocation. As the effective storage ratios of the three reservoirs converged over time, the patterns of their low water levels evolved in a largely similar manner (Figure 3).

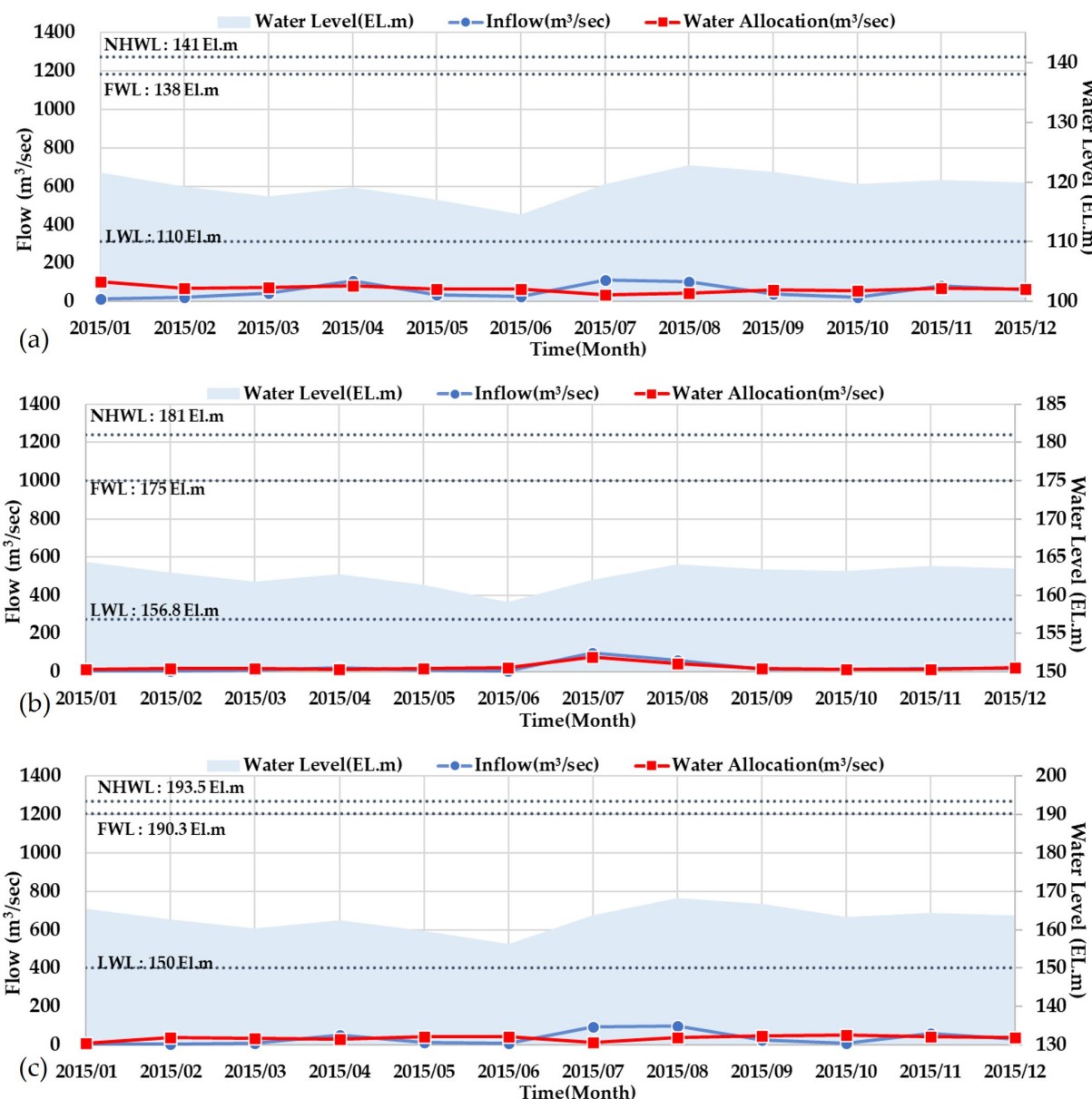

**Figure 3.** The results from (**a**) Supplementary Table S1, (**b**) Supplementary Table S2, and (**c**) Supplementary Table S3 are represented in the graphs. In this figure, the water level is depicted as a sky-blue shaded area, inflow is shown as a blue solid line, and water allocation is represented in red.

*4.2. Model Evaluation for Normal Year (2018)*

Supplementary Tables S4 (Chungju Reservoir), S5 (Soyanggang Reservoir) and S6 (Hwacheon Reservoir) show the results of applying the optimization model during the normal year of 2018, as represented in Figure 4a–c. The initial water levels (end-of-month water levels in December 2017) in December 2017 for the Chungju, Soyanggang, and Hwacheon Reservoirs were 128.47 EL.m, 180.97 EL.m, and 169.39 EL.m, respectively. Thus, the planned water supply was met for each reservoir in 2018. The lowest water levels for the three reservoirs occurred in March, recorded as 119.1 EL.m for Chungju Reservoir, 173.0 EL.m for Soyanggang Reservoir, and 165.0 EL.m for Hwacheon Reservoir. These levels were maintained to mitigate dry season impacts, and sufficient inflows were observed up to the flood season in June. The average annual inflows for 2018 were 159.7 CMS for Chungju Reservoir, 66.1 CMS for Soyanggang Reservoir, and 38.1 CMS for Hwacheon Reservoir. The average annual allocation amounts were 137.3 CMS (59.1%), 53.0 CMS (25.4%), and

34.8 CMS (15.6%). No spillway releases were required, as the maximum releases for the Chungju, Soyanggang, and Hwacheon Reservoirs were 775.0 CMS, 250.0 CMS, and 185.0 CMS, respectively. While Soyanggang Reservoir had the highest effective storage ratio at the beginning of the year, Chungju Reservoir had the highest effective storage ratio by the end of the year.

**Table 4.** Optimization model-determined water allocation and effective reservoir storage ratios for the three reservoirs in the dry year of 2015.

| 2015-MM | Model-Derived Water Allocation (CMS) | | | Model-Derived Effective Reservoir Storage Ratio (%) | | | Total Water Supply (CMS) |
|---|---|---|---|---|---|---|---|
| | Chungju | Soyanggang | Hwacheon | Chungju | Soyanggang | Hwacheon | |
| 01 | 104.3 | 8.1 | 11.6 | 26.0 | 24.7 | 24.8 | 124.0 |
| 02 | 67.6 | 40.3 | 16.1 | 19.9 | 19.9 | 20.0 | 124.0 |
| 03 | 72.6 | 36.6 | 14.8 | 15.8 | 15.8 | 15.9 | 124.0 |
| 04 | 82.1 | 29.7 | 12.2 | 19.2 | 19.2 | 19.3 | 124.0 |
| 05 | 66.9 | 43.5 | 15.9 | 14.5 | 14.5 | 14.6 | 126.3 |
| 06 | 65.1 | 44.1 | 19.5 | 8.8 | 9.1 | 7.2 | 128.7 |
| 07 | 33.7 | 12.6 | 77.8 | 20.7 | 21.6 | 16.7 | 124.0 |
| 08 | 44.4 | 37.0 | 42.7 | 29.4 | 30.5 | 23.7 | 124.0 |
| 09 | 59.8 | 46.5 | 17.8 | 26.3 | 27.4 | 21.2 | 124.0 |
| 10 | 58.6 | 53.0 | 12.5 | 20.7 | 20.7 | 20.8 | 124.0 |
| 11 | 67.8 | 44.1 | 12.1 | 22.7 | 22.7 | 22.8 | 124.0 |
| 12 | 66.7 | 37.6 | 19.7 | 21.6 | 21.6 | 21.7 | 124.0 |

**Table 5.** Optimization model-determined water allocation and effective reservoir storage ratios for the three reservoirs in the normal year of 2018.

| 2018-MM | Model-Derived Water Allocation (CMS) | | | Model-Derived Effective Reservoir Storage Ratio (%) | | | Total Water Supply (CMS) |
|---|---|---|---|---|---|---|---|
| | Chungju | Soyanggang | Hwacheon | Chungju | Soyanggang | Hwacheon | |
| 01 | 97.2 | 78.3 | 22.2 | 33.9 | 49.6 | 37.7 | 197.7 |
| 02 | 97.2 | 46.2 | 22.2 | 21.7 | 43.6 | 32.0 | 165.6 |
| 03 | 97.2 | 46.2 | 22.2 | 19.2 | 41.0 | 27.1 | 165.6 |
| 04 | 106.3 | 46.2 | 22.2 | 29.8 | 46.9 | 32.2 | 174.7 |
| 05 | 119.0 | 46.2 | 22.2 | 49.0 | 63.4 | 63.1 | 187.4 |
| 06 | 125.2 | 46.2 | 94.1 | 37.3 | 59.2 | 30.0 | 265.5 |
| 07 | 115.2 | 46.2 | 47.9 | 72.8 | 75.8 | 68.9 | 209.3 |
| 08 | 192.5 | 78.2 | 67.7 | 85.8 | 89.3 | 68.9 | 338.4 |
| 09 | 397.5 | 63.7 | 29.9 | 85.8 | 89.3 | 68.9 | 491.1 |
| 10 | 105.2 | 46.2 | 22.2 | 93.6 | 90.1 | 66.5 | 173.6 |
| 11 | 97.2 | 46.2 | 22.2 | 94.2 | 90.2 | 68.9 | 165.6 |
| 12 | 97.2 | 46.2 | 22.2 | 88.6 | 85.3 | 64.5 | 165.6 |

**Table 6.** Optimization model-determined water allocation and effective reservoir storage ratios for the three reservoirs in the wet year of 2020.

| 2020-MM | Model-Derived Water Allocation (CMS) | | | Model-Derived Effective Reservoir Storage Ratio (%) | | | Total Water Supply (CMS) |
|---|---|---|---|---|---|---|---|
| | Chungju | Soyanggang | Hwacheon | Chungju | Soyanggang | Hwacheon | |
| 01 | 97.2 | 46.2 | 22.2 | 68.0 | 49.7 | 53.3 | 165.6 |
| 02 | 97.2 | 46.2 | 22.2 | 63.1 | 46.2 | 50.3 | 165.6 |
| 03 | 97.2 | 46.2 | 22.2 | 61.1 | 44.9 | 48.1 | 165.6 |

**Table 6.** *Cont.*

| 2020-MM | Model-Derived Water Allocation (CMS) | | | Model-Derived Effective Reservoir Storage Ratio (%) | | | Total Water Supply (CMS) |
|---|---|---|---|---|---|---|---|
| | Chungju | Soyanggang | Hwacheon | Chungju | Soyanggang | Hwacheon | |
| 04 | 106.3 | 46.2 | 22.2 | 52.9 | 42.9 | 44.1 | 174.7 |
| 05 | 119.0 | 46.2 | 31.1 | 49.6 | 47.3 | 47.4 | 196.3 |
| 06 | 125.2 | 46.2 | 22.2 | 35.7 | 44.4 | 44.9 | 193.6 |
| 07 | 115.2 | 46.2 | 26.5 | 68.1 | 61.0 | 47.2 | 187.9 |
| 08 | 778.0 | 250.0 | 185.0 | 85.8 | 89.3 | 68.9 | 1213.0 |
| 09 | 458.8 | 250.0 | 185.0 | 85.8 | 89.3 | 68.9 | 893.8 |
| 10 | 105.2 | 46.2 | 22.2 | 77.0 | 85.0 | 65.4 | 173.6 |
| 11 | 97.2 | 46.2 | 22.2 | 67.0 | 80.5 | 61.8 | 165.6 |
| 12 | 97.2 | 46.2 | 24.4 | 54.9 | 74.4 | 55.0 | 167.8 |

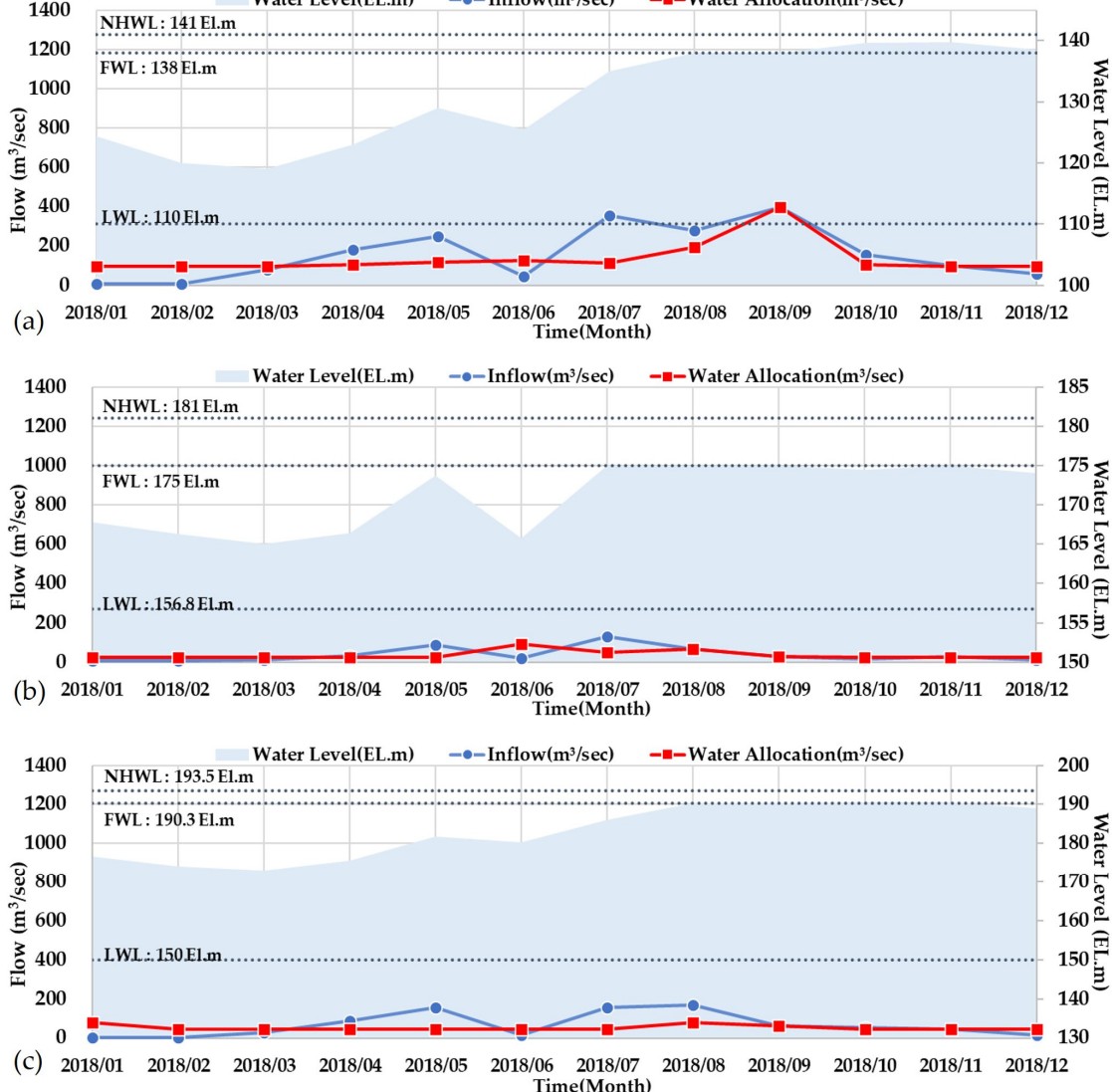

**Figure 4.** The results from (**a**) Supplementary Table S4, (**b**) Supplementary Table S5, and (**c**) Supplementary Table S6 are represented in the graphs.

*4.3. Model Evaluation for Wet Year (2020)*

Supplementary Tables S7 (Chungju Reservoir), S8 (Soyanggang Reservoir) and S9 (Hwacheon Reservoir) present the results during the wet year of 2020, as illustrated in

Figure 5a–c. South Korea experienced extreme floods in 2020. In 2019, the Han River Basin experienced lower-than-normal inflows, leading to minimal releases from the three reservoirs due to concerns about a prolonged drought. Consequently, water levels were high at the end of 2019. The initial water levels (December 2019 month-end water levels) for the Chungju, Soyanggang, and Hwacheon Reservoirs were 134.4 EL.m, 176.6 EL.m, and 171.6 EL.m, respectively. In 2020, reservoirs had the planned water supply for each reservoir and the spillway releases. The average annual inflows for the Chungju, Soyanggang, and Hwacheon Reservoirs in 2020 were 192.6 CMS, 106.4 CMS, and 102.3 CMS, respectively. The average annual allocation amounts were 191.1 CMS (60.1%), 80.2 CMS (26.2%), and 49.3 CMS (13.7%). Although significant flooding occurred in 2020, a substantial drop in inflow occurred after the flood season, resulting in a decrease in the effective storage ratio.

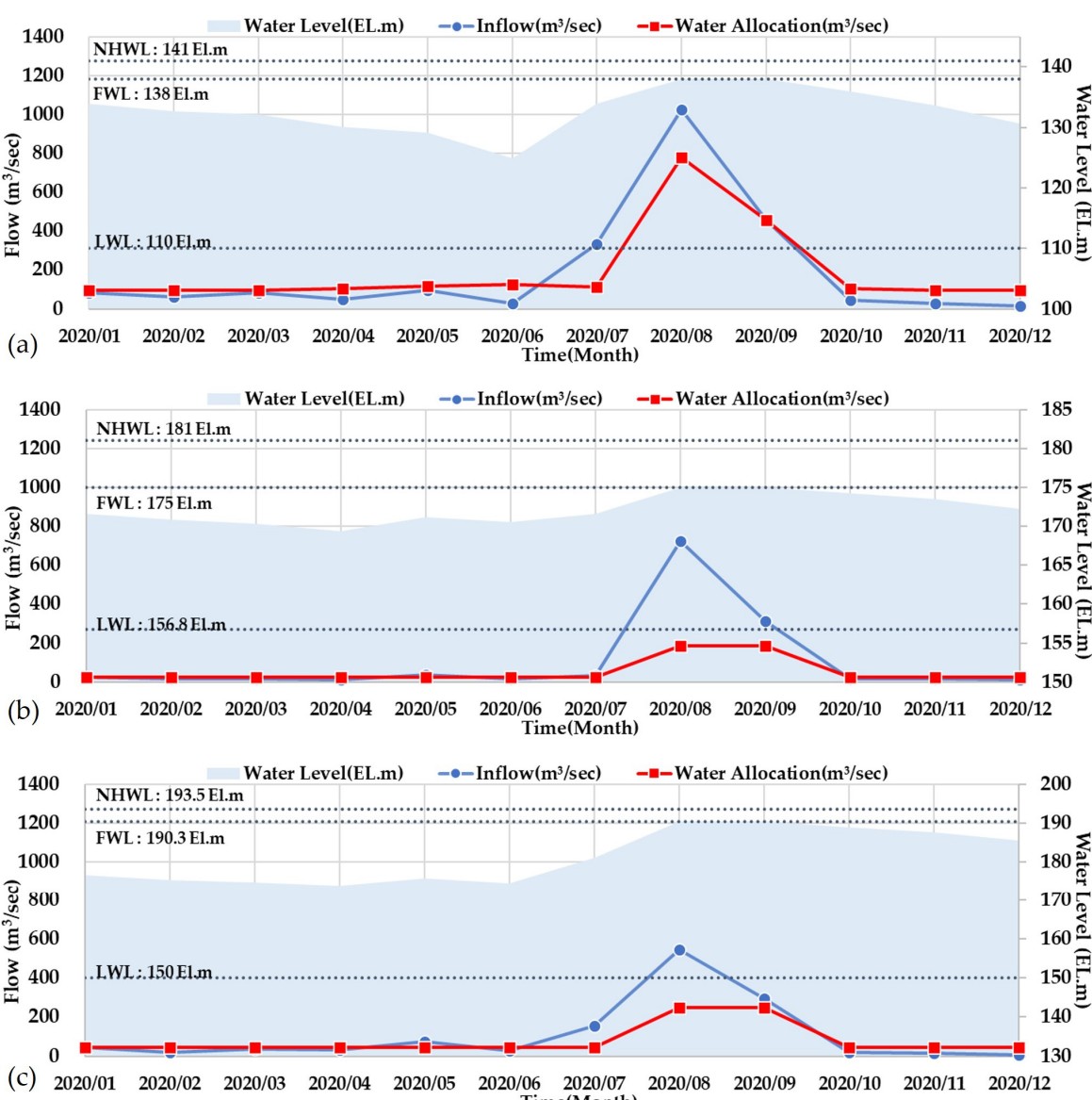

**Figure 5.** The results from (**a**) Supplementary Table S7, (**b**) Supplementary Table S8, and (**c**) Supplementary Table S9 are represented in the graphs.

## 5. Discussion

Globally, constructing new reservoirs is becoming increasingly challenging, necessitating the exploration of methods to repurpose existing reservoirs for multiple uses. In South Korea, water supply functions have been newly added to existing hydropower reservoirs.

This approach is expected to contribute to the country's sustainable development goals. The following subsections delve into the specifics of optimization and performance assessment under varying hydrological conditions. We also engage in a comparative analysis with prior research in the field, further establishing the relevance and contributions of the present study to the literature on reservoir operation.

*5.1. Optimization and Performance Assessment in Dry Year (2015)*

Table 4 presents the optimized water allocation and effective storage ratios for the three reservoirs during the dry year of 2015. Table 7 compares the monthly water allocation based on historical data with optimization results for three reservoirs in the dry year. Figure 6a depicts the optimized water supply of 2015. Figure 7a displays the historical monthly water allocation ratios for the three reservoirs, while Figure 7b illustrates the corresponding ratios determined with the optimization model.

**Table 7.** Comparison of historical data and optimization results for monthly water allocation in three reservoirs for the dry year of 2015.

| | Historical Data | | | | | | | | Optimization Results | | | | | | | |
|---|---|---|---|---|---|---|---|---|---|---|---|---|---|---|---|---|
| | Chungju Reservoir | | Soyanggang Reservoir | | Hwacheon Reservoir | | Total | | Chungju Reservoir | | Soyanggang Reservoir | | Hwacheon Reservoir | | Total | |
| 2015-MM | CMS | % | CMS | % | CMS | % | CMS | % | CMS | % | CMS | % | CMS | % | CMS | % |
| 01 | 79.1 | 59.0 | 45.1 | 33.6 | 9.9 | 7.4 | 134.0 | 100 | 104.3 | 84.2 | 8.1 | 6.5 | 11.6 | 9.3 | 124.0 | 100 |
| 02 | 78.9 | 8.9 | 44.9 | 33.5 | 10.2 | 7.6 | 134.1 | 100 | 67.6 | 54.5 | 40.3 | 32.5 | 16.1 | 13.0 | 124.0 | 100 |
| 03 | 85.8 | 67.6 | 33.1 | 26.1 | 8.1 | 6.4 | 127.1 | 100 | 72.6 | 58.5 | 36.6 | 29.5 | 14.8 | 12.0 | 124.0 | 100 |
| 04 | 85.6 | 67.7 | 28.2 | 22.3 | 12.7 | 10.0 | 126.5 | 100 | 82.1 | 66.2 | 29.7 | 24.0 | 12.2 | 9.9 | 124.0 | 100 |
| 05 | 85.1 | 53.7 | 41.5 | 26.1 | 32.1 | 20.2 | 158.6 | 100 | 66.9 | 53.0 | 43.5 | 34.4 | 15.9 | 12.6 | 126.3 | 100 |
| 06 | 32.7 | 31.3 | 44.3 | 42.3 | 27.6 | 26.4 | 104.6 | 100 | 65.1 | 50.6 | 44.1 | 34.3 | 19.5 | 15.1 | 128.7 | 100 |
| 07 | 16.5 | 28.0 | 4.8 | 8.2 | 37.5 | 63.8 | 58.8 | 100 | 33.7 | 27.2 | 12.6 | 10.2 | 77.8 | 62.7 | 124.0 | 100 |
| 08 | 25.1 | 52.1 | 4.9 | 10.2 | 18.2 | 37.7 | 48.2 | 100 | 44.4 | 35.8 | 37.0 | 29.8 | 42.7 | 34.4 | 124.0 | 100 |
| 09 | 29.0 | 29.5 | 4.7 | 4.7 | 64.6 | 65.8 | 98.3 | 100 | 59.8 | 48.2 | 46.5 | 37.5 | 17.8 | 14.3 | 124.0 | 100 |
| 10 | 29.7 | 30.0 | 36.1 | 36.4 | 33.3 | 33.6 | 99.1 | 100 | 58.6 | 47.2 | 53.0 | 42.7 | 12.5 | 10.1 | 124.0 | 100 |
| 11 | 16.4 | 56.9 | 11.9 | 41.5 | 0.5 | 1.6 | 28.7 | 100 | 67.8 | 54.7 | 44.1 | 35.6 | 12.1 | 9.8 | 124.0 | 100 |
| 12 | 14.6 | 60.7 | 4.9 | 20.4 | 4.5 | 18.9 | 24.0 | 100 | 66.7 | 53.8 | 37.6 | 30.4 | 19.7 | 15.9 | 124.0 | 100 |
| average | 48.2 | 49.6 | 25.4 | 25.5 | 21.6 | 25.0 | 95.2 | 100 | 65.8 | 52.8 | 36.1 | 29.0 | 22.7 | 18.3 | 124.6 | 100 |

In a dry year like 2015 with low initial water levels and modest inflows, it is challenging to satisfy the planned water supply. Thus, operations focused on maintaining at least the instream flow while optimizing each reservoir's effective storage ratio. Thus, the storage ratios of all three reservoirs converged, and their total supply exceeded Paldang Reservoir's required discharge. Given that the reservoirs are managed by different agencies, adopting a balanced approach based on an effective storage ratio method is practical, especially under basin-wide severe drought conditions. In periods of low inflow, it is observed that optimal results were well-achieved in accordance with the objective, as there was no occasion for the reservoir levels to reach the normal high water level, unlike in normal or flood seasons. Slight differences occurred in the monthly effective storage ratios of the three reservoirs, influenced by the watershed-specific inflow rates. In June 2015, the effective storage ratios dropped below 10%, marking a critical state; however, the subsequent reservoir operation raised low water levels across all three reservoirs. Notably, the combined release from the three reservoirs was identical to the requirements of the Paldang Reservoir's discharge. By exceeding instream flows, the model aimed to increase storage ratios while minimizing releases from the reservoir with the lowest storage ratio, thereby achieving comparable effective storage ratios over time.

According to the actual and historical operations, water was mainly supplied from the Chungju and Soyanggang Reservoirs, as the water supply function was inactivated in Hwacheon Reservoir in 2015. Except for the period from January to May, the supply from the Chungju and Soyanggang Reservoirs was not sufficient to satisfy the required discharge of Paldang Reservoir due to the continuous drought. After June 2015, the combined discharge of the three reservoirs was smaller than Paldang Reservoir's required discharge.

During the months from July to October 2015, the Hwacheon Reservoir released more than the other two reservoirs, thereby assisting with water supply for the Seoul area.

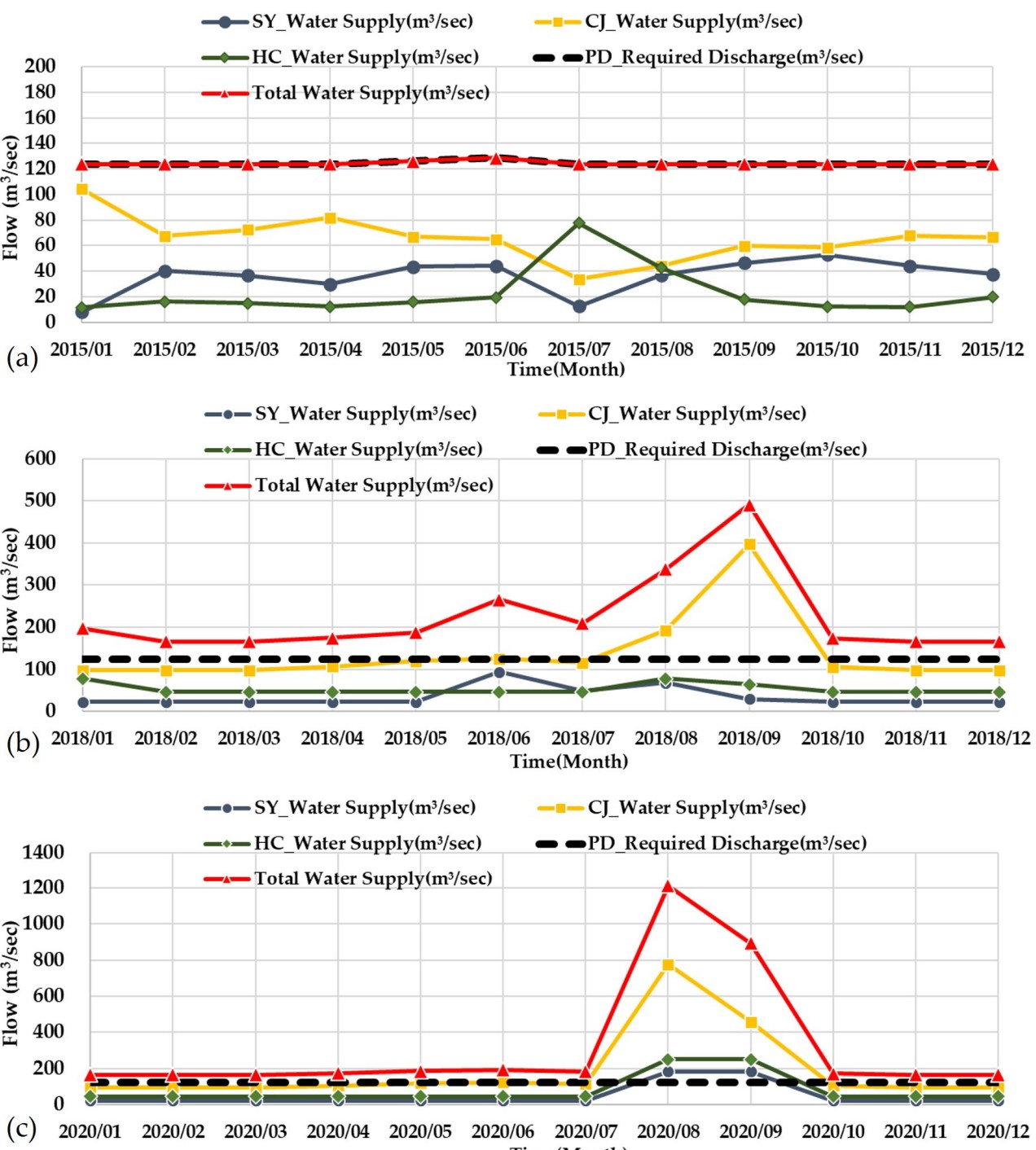

**Figure 6.** Optimized water supply for the years (**a**) 2015, (**b**) 2018, and (**c**) 2020 in correspondence with Tables 4–6.

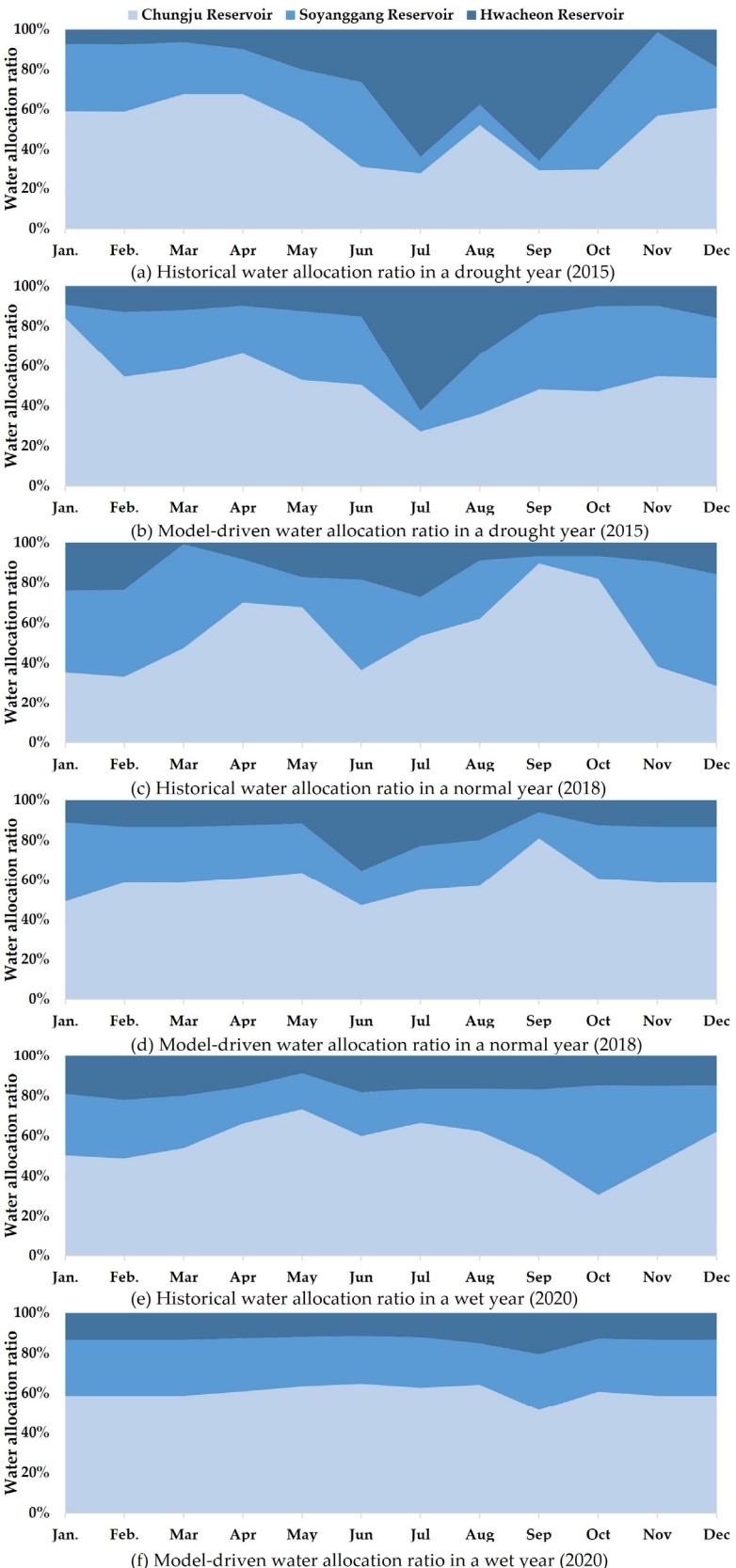

**Figure 7.** Subfigures (**a,c,e**) presenting the historical data on monthly water allocation ratios for three reservoirs in the years 2015, 2018, and 2020, respectively. Subfigures (**b,d,f**) displaying the monthly water allocation ratios for these reservoirs in the same years, as determined with the optimization model.

According to the optimization model results, the total allocation amount of the three reservoirs could supply the required discharge of each reservoir through the joint operation. Additionally, there was a tendency for stable allocation among the three reservoirs consistently during the dry season, as shown by a significant reduction in differences in the water allocation ratios of the three reservoirs (Figure 7a,b).

*5.2. Optimization and Performance Assessment in Normal Year (2018)*

Table 5 shows the optimized water allocation and effective storage ratios for the three reservoirs in the normal year. Table 8 compares the monthly water allocation based on historical data with optimization results for the three reservoirs in the normal year. Figure 6b shows the optimized water supply for the normal year. Figure 7c presents the historical data on monthly water allocation ratios for the three reservoirs, while Figure 7d provides the corresponding ratios estimated with the optimization model.

**Table 8.** Comparison of historical data and optimization results for monthly water allocation in three reservoirs for the normal year of 2018.

| | Historical Data | | | | | | | | Optimization Results | | | | | | | |
|---|---|---|---|---|---|---|---|---|---|---|---|---|---|---|---|---|
| **2018-MM** | **Chungju Reservoir** | | **Soyanggang Reservoir** | | **Hwacheon Reservoir** | | **Total** | | **Chungju Reservoir** | | **Soyanggang Reservoir** | | **Hwacheon Reservoir** | | **Total** | |
| | **CMS** | **%** | **CMS** | **%** | **CMS** | **%** | **CMS** | **%** | **CMS** | **%** | **CMS** | **%** | **CMS** | **%** | **CMS** | **%** |
| 01 | 52.0 | 35.3 | 60.0 | 40.8 | 35.2 | 23.9 | 147.2 | 100 | 97.2 | 49.2 | 78.3 | 39.6 | 22.2 | 11.2 | 197.7 | 100 |
| 02 | 52.0 | 33.1 | 68.1 | 43.3 | 37.0 | 23.6 | 157.1 | 100 | 97.2 | 58.7 | 46.2 | 27.9 | 22.2 | 13.4 | 165.6 | 100 |
| 03 | 52.0 | 47.5 | 56.6 | 51.7 | 0.9 | 0.9 | 109.5 | 100 | 97.2 | 58.7 | 46.2 | 27.9 | 22.2 | 13.4 | 165.6 | 100 |
| 04 | 107.1 | 70.1 | 32.9 | 21.5 | 12.8 | 8.4 | 152.7 | 100 | 106.3 | 60.9 | 46.2 | 26.5 | 22.2 | 12.7 | 174.7 | 100 |
| 05 | 343.3 | 67.8 | 75.5 | 14.9 | 87.3 | 17.3 | 506.1 | 100 | 119.0 | 63.5 | 46.2 | 24.7 | 22.2 | 11.9 | 187.4 | 100 |
| 06 | 128.5 | 36.5 | 158.9 | 45.2 | 64.5 | 18.3 | 351.9 | 100 | 125.2 | 47.2 | 46.2 | 17.4 | 94.1 | 35.5 | 265.5 | 100 |
| 07 | 177.3 | 53.4 | 64.4 | 19.4 | 90.3 | 27.2 | 331.9 | 100 | 115.2 | 55.0 | 46.2 | 22.1 | 47.9 | 22.9 | 209.3 | 100 |
| 08 | 99.1 | 62.0 | 46.4 | 29.1 | 14.2 | 8.9 | 159.7 | 100 | 192.5 | 56.9 | 78.2 | 23.1 | 67.7 | 20.0 | 338.4 | 100 |
| 09 | 284.7 | 89.7 | 11.3 | 3.6 | 21.5 | 6.8 | 317.4 | 100 | 397.5 | 80.9 | 63.7 | 13.0 | 29.9 | 6.1 | 491.1 | 100 |
| 10 | 118.2 | 82.0 | 16.3 | 11.3 | 9.6 | 6.7 | 144.1 | 100 | 105.2 | 60.6 | 46.2 | 26.6 | 22.2 | 12.8 | 173.6 | 100 |
| 11 | 57.7 | 38.3 | 78.4 | 52.1 | 14.6 | 9.7 | 150.6 | 100 | 97.2 | 58.7 | 46.2 | 27.9 | 22.2 | 13.4 | 165.6 | 100 |
| 12 | 51.8 | 28.6 | 101.0 | 55.7 | 28.6 | 15.8 | 181.4 | 100 | 97.2 | 58.7 | 46.2 | 27.9 | 22.2 | 13.4 | 165.6 | 100 |
| average | 52.0 | 35.3 | 60.0 | 40.8 | 35.2 | 23.9 | 147.2 | 100 | 137.3 | 59.1 | 53.0 | 25.4 | 34.8 | 15.6 | 225.0 | 100 |

During a normal year, a reservoir releases water based on the planned water supply. However, the model aimed to increase the lowest effective storage ratio for the three reservoirs. Spillways opened throughout the flood season to maintain reliable reservoir operation. The reservoir operation model focused on the dry season rather than the flood season, focusing on the optimization model as it aimed to build a long-term reservoir operation plan. The model executed spillway releases during the flood season to enhance the effective storage ratio during the next dry season. Each reservoir exceeded its planned water supply, and at the same time, the sum of the three reservoir releases surpassed Paldang Reservoir's required discharge (Table 5, Figure 6b). Due to different inflows and planned water supplies across the three reservoirs, a uniform storage ratio was more challenging to achieve. By implementing reservoir operation strategies aimed at increasing the water level in the reservoir with the lowest effective storage ratio, the differences in effective storage ratios were substantially reduced. The model aimed to raise the water level of the reservoir with the minimum water level considering the water availability. This dropped the differences in the effective storage ratio from over 15% to approximately less than 1%.

According to the actual and historical operations, in 2018 right after the extreme drought spanning from 2014 to 2017, the Chungju and Soyanggang Reservoirs could not meet their planned water supply. It was only after incorporating the water allocation from Hwacheon Reservoir that the combined allocation exceeded the anticipated discharge of the Paldang Reservoir. The Hwacheon Reservoir, still functioning as hydropower generation in 2018, had release fluctuations in monthly water allocation during the dry season, which lasted from October to May.

The optimization model released more water than the monthly planned water supply, thereby ensuring that the sum of water allocations from all three reservoirs satisfied the discharge requirements of the Paldang Reservoir. This setup resulted in uniform water allocation ratios among the three reservoirs during the dry season (Figure 7c,d).

*5.3. Optimization and Performance Assessment in Wet Year (2020)*

Table 6 outlines the optimized water allocation and effective storage ratios for the three reservoirs in the wet year. Table 9 compares the monthly water allocation based on historical data with optimization results for the three reservoirs in the wet year. Figure 6c shows the optimized water supply. Figure 7e presents the historical data on monthly water allocation ratios for the three reservoirs, while Figure 7f provides the corresponding ratios as estimated with the optimization model.

**Table 9.** Comparison of historical data and optimization results for monthly water allocation in three reservoirs for the wet year of 2020.

| | Historical Data | | | | | | | | Optimization Results | | | | | | | |
|---|---|---|---|---|---|---|---|---|---|---|---|---|---|---|---|---|
| **2020-MM** | **Chungju Reservoir** | | **Soyanggang Reservoir** | | **Hwacheon Reservoir** | | **Total** | | **Chungju Reservoir** | | **Soyanggang Reservoir** | | **Hwacheon Reservoir** | | **Total** | |
| | **CMS** | **%** | **CMS** | **%** | **CMS** | **%** | **CMS** | **%** | **CMS** | **%** | **CMS** | **%** | **CMS** | **%** | **CMS** | **%** |
| 01 | 74.1 | 50.2 | 45.3 | 30.7 | 28.2 | 19.1 | 147.6 | 100 | 97.2 | 58.7 | 46.2 | 27.9 | 22.2 | 13.4 | 165.6 | 100 |
| 02 | 74.7 | 48.8 | 44.4 | 29.0 | 33.9 | 22.2 | 153.0 | 100 | 97.2 | 58.7 | 46.2 | 27.9 | 22.2 | 13.4 | 165.6 | 100 |
| 03 | 75.0 | 53.9 | 36.3 | 26.1 | 27.9 | 20.0 | 139.2 | 100 | 97.2 | 58.7 | 46.2 | 27.9 | 22.2 | 13.4 | 165.6 | 100 |
| 04 | 107.6 | 66.1 | 29.7 | 18.2 | 25.7 | 15.7 | 162.9 | 100 | 106.3 | 60.9 | 46.2 | 26.5 | 22.2 | 12.7 | 174.7 | 100 |
| 05 | 121.7 | 73.3 | 29.5 | 17.8 | 14.8 | 8.9 | 166.0 | 100 | 119.0 | 63.5 | 46.2 | 24.7 | 22.2 | 11.9 | 187.4 | 100 |
| 06 | 109.6 | 60.0 | 39.7 | 21.7 | 33.5 | 18.3 | 182.8 | 100 | 125.2 | 64.7 | 46.2 | 23.9 | 22.2 | 11.5 | 193.6 | 100 |
| 07 | 124.3 | 66.5 | 31.8 | 17.0 | 30.8 | 16.5 | 186.8 | 100 | 115.2 | 62.8 | 46.2 | 25.2 | 22.2 | 12.1 | 183.6 | 100 |
| 08 | 541.7 | 62.3 | 185.1 | 21.3 | 143.3 | 16.5 | 870.0 | 100 | 778.0 | 64.1 | 250.0 | 20.6 | 185.0 | 15.3 | 1213 | 100 |
| 09 | 286.5 | 49.3 | 196.4 | 33.8 | 98.2 | 16.9 | 581.2 | 100 | 458.0 | 51.3 | 250.0 | 28.0 | 185.0 | 20.7 | 893.8 | 100 |
| 10 | 46.4 | 30.5 | 83.2 | 54.7 | 22.7 | 14.9 | 152.3 | 100 | 105.2 | 60.6 | 46.2 | 26.6 | 22.2 | 12.8 | 173.6 | 100 |
| 11 | 69.6 | 46.3 | 58.3 | 38.7 | 22.6 | 15.0 | 150.5 | 100 | 97.2 | 58.7 | 46.2 | 27.9 | 22.2 | 13.4 | 165.6 | 100 |
| 12 | 94.9 | 62.0 | 35.4 | 23.2 | 22.7 | 14.8 | 152.9 | 100 | 97.2 | 58.7 | 46.2 | 27.9 | 22.2 | 13.4 | 165.6 | 100 |
| average | 74.1 | 50.2 | 45.3 | 30.7 | 28.2 | 19.1 | 147.6 | 100 | 191.1 | 60.1 | 80.2 | 26.2 | 49.3 | 13.7 | 320.6 | 100 |

In a wet year, water releases are based on the planned water supply of each reservoir, like the normal year, and at the same time, the lowest effective storage ratios of the three reservoirs are increased. To ensure reliable reservoir operation, the reservoirs opened spillways and released water throughout the flood season. Notably, these spillway releases were more substantial in wet years compared to normal or dry years. To establish a long-term reservoir operation plan, the developed optimization model was primarily focused on operation for the dry season rather than for the flood season. In the dry season, the reservoirs released the minimum required supply to improve the effective storage ratio. Each reservoir released water above its planned water supply, and the sum of supplies from the three reservoirs exceeded the required discharge of the Paldang Reservoir (Table 6, Figure 6c). The operations aimed to maximize the storage ratio of each reservoir, similar to the approach in the normal year of 2018. In 2020, exceeding the normal inflows from January to the flood season led to greater water supply, hydropower generation, and spillway releases. The high monthly inflows resulted in significant fluctuations in the effective storage ratio throughout 2020.

Hwacheon Reservoir commenced its pilot operation in April 2020. Hwacheon Reservoir has consistently released 22.2 CMS from April to the present, leading to a more balanced water allocation ratio for all three reservoirs. However, there are instances, such as in October 2020, when the allocation ratio of Soyanggang Reservoir surpasses that of Chungju Reservoir. This happens because water releases are based on individual reservoir standards rather than a system operation among the three reservoirs. However, the model results ensured a relatively stable water allocation ratio by considering the storage ratios of all three reservoirs in determining the allocation amount (Figure 7e,f).

*5.4. Comparative Analysis with Previous Studies in Reservoir Operation*

To contextualize the contributions of the present study, it is instructive to compare its methodologies and findings with those of previous studies in the field of reservoir operation and water allocation. One study used two-stage stochastic linear programming to optimize reservoir operations in the Han River Basin, focusing solely on the multipurpose reservoirs of Chungju and Soyanggang [48]. In the first stage, the model determined reservoir storage, while in the second stage, it set water supply and environmental flow rates based on actual demand. The optimization aimed to minimize discrepancies between the target and actual reservoir storage and any water supply and environmental flow shortages. The study also utilizes a Hedging Rule to adjust planned release rates based on current reservoir storage, thereby enhancing the model's applicability and relevance.

In contrast to this study, which relied on an artificial Hedging Rule, our research formulated an optimal water allocation model that incorporated the effective storage ratio of each reservoir into the allocation calculations. This approach ensured a more balanced and efficient distribution of downstream water requirements. We also broadened the scope by incorporating a transformed hydropower reservoir with an add-on water supply function and two multipurpose reservoirs. This offered a versatile and comprehensive solution for reservoir management, particularly in regions where collaboration between different types of reservoirs is essential for optimal performance.

Another study focused on the coordinated operation of multipurpose reservoirs (Soyanggang, Chungju, and Hoengseong), a water supply-only reservoir (Gwangdong), and a large-scale hydropower reservoir (Hwacheon) within the Han River Basin [49]. The paper employed a five-level Hedging Rule and used mixed integer linear programming (MILP) to develop a reservoir operation model. The model aimed to approximate the actual storage volume of existing reservoirs to target storage volumes while maximizing water supply rates and ensuring maximum river maintenance flows. The paper focused on phased water supply reductions to optimize water supply during drought conditions.

As opposed to the approach taken in this research, which employed a five-level Hedging Rule and MILP for optimizing water allocation primarily during drought conditions, our research focused on a broader range of hydrological scenarios. Our research aimed for efficient water allocation from the Soyanggang, Chungju, and Hwacheon Reservoirs, which are responsible for water supply in the metropolitan area. Additionally, our work introduced a transformed hydropower reservoir into the system, offering a more versatile and sustainable solution for water resources management.

## 6. Conclusions

In summary, the proposed method optimized water supply capacity across multiple upstream and downstream reservoirs and focused on rational operational strategies for downstream water requirements. Validated against historical inflow data, the model effectively balanced the unique operational needs of upstream hydropower and other multipurpose reservoirs while ensuring optimal downstream water supply.

When comparing the monthly storage ratios of each reservoir, the model aimed to minimize differences in storage ratios while determining the allocation. During the dry year (2015), the effective storage ratios of the three reservoirs were operated almost equally each month. The actual annual average allocation ratios for the Chungju, Soyanggang, and Hwacheon Reservoirs were 49.6%, 25.5%, and 25.0%, respectively, while the model-estimated ratios were 52.8%, 29.0%, and 18.3%. The higher allocation for the Chungju and Soyanggang Reservoirs compared to actual data was due to their higher effective storage ratios and greater inflows. In the normal year (2018), the actual annual average allocation ratios for the Chungju, Soyanggang, and Hwacheon Reservoirs were 35.3%, 40.8%, and 23.9%, respectively. Soyanggang Reservoir had more discharge than Chungju Reservoir because there were periods when Chungju Reservoir did not supply the basic planned amount. However, considering the effective storage ratio and the planned water supply, the estimated allocation ratios changed to 59.1%, 25.4%, and 15.6%, respectively, increasing the

proportion allocated to Chungju Reservoir. In the wet year (2020), the actual annual average allocation ratios for the Chungju, Soyanggang, and Hwacheon Reservoirs, respectively, were 50.2%, 30.7%, and 19.1%, while the model-estimated ratios were 60.1%, 26.2%, and 13.7%. The Chungju Reservoir, with its larger catchment area and higher inflow, had a higher estimated proportion of the allocation.

This study delivers several key findings that significantly contribute to the field of water resources management. First, the optimization model effectively allocated water among three critical reservoirs—Chungju, Soyanggang, and Hwacheon—across diverse hydrological scenarios, including dry, normal, and wet years. It should be noted that, in addition to the case studies presented for these specific hydrological conditions, the model was also rigorously tested for every single year from 2004 to 2022, yielding consistently satisfactory results. This extensive validation across a wide range of inflow conditions underscores the model's robustness and reliability. Second, the model adeptly balanced effective storage ratios among these reservoirs, enhancing drought preparedness and water resource management. Third, historical data served as a robust validation mechanism, confirming the model's effectiveness in appropriately distributing downstream water supply. Specifically, during a dry year such as 2015, the model excelled in water management even under low inflow conditions. In a normal year like 2018, the model not only maintained the storage ratios for each reservoir but also exceeded the required discharge for Paldang Reservoir. Additionally, the model demonstrated adaptability in a wet year, like 2020, by effectively managing increased inflows. The pilot operation of Hwacheon Reservoir further solidified the model's capability, resulting in more balanced water allocation across the reservoirs. Moreover, the model was multiobjective, serving various functions, such as maintaining instream flows and preparing for extreme hydrological events. These key findings not only have academic implications but also offer practical utility in real-world reservoir and water resources management.

While the study's results underscore the model's efficiency in water allocation and flood control, it is crucial to acknowledge the inherent trade-offs in reservoir operation. Environmental impacts can be mitigated through ongoing monitoring and adaptive management strategies, while stakeholder engagement can help resolve social conflicts. By recognizing and proactively managing these trade-offs, the proposed model aligns with broader sustainability objectives. This integrated approach contributes significantly to the sustainable operation of reservoir systems, thus benefiting both the environment and the communities they serve.

This study presents a comprehensive model for reservoir management but has limitations and uncertainties. The model's scope was restricted by its exclusion of various hydrological factors, like tributary inflows and water losses, and by assumptions regarding discharge capacities of the reservoirs involved. While validated against historical data, the model was not tested in real-world settings and did not extend to extreme hydrological scenarios. One limitation of our model is that it did not address the potential conflicts between water supply and hydropower generation. Specifically, the optimal timing for each may differ, and adding a water supply function to hydropower reservoirs could adversely affect hydropower output. This impact was not examined in the current study. The limitations and uncertainties of the proposed methodology include the following: The optimization model was designed to meet the objective function within specified constraints and may yield infeasible solutions when not all conditions can be satisfied. This is particularly true during extreme drought conditions, where the planned water supply may be insufficient. To address this, we modified the constraints to maintain instream flow during droughts. Additionally, the use of linear-programming techniques limited the model's applicability to nonlinear equations, which is why hydropower generation was excluded from this study.

While the current study provided model results based on predefined scenarios such as dry, normal, and wet years, it is crucial to highlight the model's flexibility in allowing allocation amounts to be estimated based on user-defined inflow periods. Future research perspectives include capitalizing on this flexibility by testing the model's adaptability and

performance under various inflow conditions. Another significant aspect to consider is the integration of other hydrological factors. The model was designed with the assumption that the sum of the allocations from the three reservoirs would exceed each reservoir's required discharge. However, it did not account for additional factors, like tributary inflows, water withdrawals, losses, and return flows. Given that the Han River system is South Korea's largest and maintains higher inflow levels even during extended droughts, it is imperative for future research to integrate the allocation model with a water balance model for more precise inflow estimates into the Paldang Reservoir. To further improve model performance, upcoming research should focus on developing a simulation model using the rule curves of the three reservoirs and advanced reservoir allocation techniques. Such an approach is anticipated to yield a more realistic operational plan closely aligned with real-world conditions. The optimally generated plan could then be incorporated into this new simulation model. Lastly, future studies could also benefit from generating long-term inflow data to evaluate the model's robustness over extended timeframes. This would enable the model to be tested against more severe drought and flood scenarios, providing valuable insights into its long-term applicability for water resources management.

In summary, the model presented in this study offered a robust framework for the rational operation of three reservoirs, Chungju, Soyanggang, and Hwacheon, which are crucial for the water supply in Seoul. By enabling these reservoirs to be operated at comparable storage ratios, the model can serve as a valuable guide for proposing rational operational plans to stakeholders. Furthermore, its versatility allows for the examination of long-term preparedness strategies against extreme events, like droughts and floods, through preliminary analyses. This is especially pertinent during dry years when water supply is highly vulnerable, making the model an objective indicator in such scenarios. Importantly, the model not only considers the storage ratio between reservoirs but also takes into account the planned water supply for each, delivering optimal outcomes for both joint and individual operations. These findings underline the effectiveness of the proposed method in achieving a balanced and reliable water supply, and they offer valuable insights for future research and practical applications in water resources management.

**Supplementary Materials:** The following supporting information can be downloaded at: https://www.mdpi.com/article/10.3390/w15203555/s1; Table S1. Optimization results for Chungju Reservoir during drought year based on initial water level of 126.2 EL.m; Table S2. Optimization results for Soyanggang Reservoir during drought year based on initial water level of 165.2 EL.m; Table S3. Optimization results for Hwacheon Reservoir during drought year based on initial water level of 165.8 EL.m; Table S4. Optimization results for Chungju Reservoir during drought year based on initial water level of 128.5 EL.m; Table S5. Optimization results for Soyanggang Reservoir during drought year based on initial water level of 181.0 EL.m; Table S6. Optimization results for Hwacheon Reservoir during drought year based on initial water level of 169.4 EL.m; Table S7. Optimization results for Chungju Reservoir during drought year based on initial water level of 134.4 EL.m; Table S8. Optimization results for Soyanggang Reservoir during drought year based on initial water level of 176.6 EL.m; Table S9. Optimization results for Hwacheon Reservoir during drought year based on initial water level of 171.6 EL.m.

**Author Contributions:** Conceptualization, E.L. and J.Y. (Jaeeung Yi); methodology, E.L. and J.Y. (Jaeeung Yi); software, E.L.; validation, E.L.; formal analysis, E.L.; investigation, E.L.; resources, E.L., J.J., S.L., J.Y. (Jeongin Yoon), S.Y. and J.Y. (Jaeeung Yi); data curation, E.L.; writing—original draft preparation, E.L., J.J. and J.Y. (Jaeeung Yi); writing—review and editing, S.Y. and J.Y. (Jaeeung Yi); visualization, E.L.; supervision, J.Y. (Jaeeung Yi); project administration, E.L. and J.Y. (Jaeeung Yi); funding acquisition, J.Y. (Jaeeung Yi). All authors have read and agreed to the published version of the manuscript.

**Funding:** This work was supported by the Korea Environment Industry & Technology Institute (KEITI) through the Water Management Project for Drought funded by the Korea Ministry of Environment (MOE) (2022003610004).

**Data Availability Statement:** Not applicable.

**Conflicts of Interest:** The authors declare no conflict of interest.

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
