# Peer review of "Development of an Optimal Water Allocation Model for Reservoir System Operation"

_water, doi:10.3390/w15203555_

Round 1

Reviewer 1 Report

The article addresses the key problem of optimizing water resources management. This is particularly relevant in the context of the observed climate change. I recommend the article for publication after taking into account the following comments.

Abstract should include reference to specific numerical values resulting from the conducted research.

Please provide more details on the water management (seasonal cycle of dam operation) carried out at Imnam Reservoir. In my opinion, this is key information for the amount of water flowing into the first of the cascade of all reservoirs.

Some of the tables included in the Results chapter, please include as supplements.

Chapter discussion in scientific papers is to confront (or in composing) your own results to previous similar studies. Unfortunately, I do not find that here (Line 479-625). Not a single publication is cited and the content of this chapter itself is really a description of the results.

 The Conclusion chapter is too broad. It should provide a clear synthesis of the results obtained, the limitations of the proposed methodology, and further research perspectives.

Reviewer 2 Report

Interesting paper and modeling approach but I fail to see it being a contribution to the extensive literature on reservoir operation.   See my comments in the paper.

Reviewer 3 Report

Thanks for inviting me to review the manuscript entitled “Development of an optimal water allocation model for reservoirs system operation.” Although the manuscript has been written well, however, it would be good if the authors could incorporate the below comments in the text.

Now a day the large-scale water reservoir/dam is not considered as a green infrastructure and development. The authors have also mentioned the social conflicts of creating a dam in the introduction section. Therefore, the authors should refer to some examples from around the world to mitigate the impacts of creating dams on society and biodiversity. Most importantly to establish the necessity of operating a dam at the cost of what (i.e., what is the trade-off) should be explained in the introduction section and a resolution in the conclusion section.

The authors should also explain if there is any uncertainty in the proposed models for the allocation/distribution of water in the methods section. The discussion section should also explain the multi-purpose use of dams (e.g., hydropower, irrigation, etc.) so that it can contribute to the sustainable development of the country.     

Round 2

Reviewer 2 Report

The literature is full of papers discussing multipurpose multi-reservoir system operation - including hydropower and water supply.   I suggest reviewing some of them to get a better idea on how to improve your model descriptions.  

Reviewer 3 Report

Thanks for addressing the comments.